

# Three-layer heterogeneous network based on the integration of CircRNA information for MiRNA-disease association prediction

Jia Qu[1], Shuting Liu[1], Han Li[1], Jie Zhou[2], Zekang Bian[3], Zihao Song[1] and Zhibin Jiang[2]

[1] Changzhou University, School of Computer Science and Artificial Intelligence, Changzhou, Jiangsu, China
[2] Shaoxing University, School of Computer Science and Engineering, Shaoxing, Zhejiang, China
[3] Jiangnan University, School of AI & Computer Science, Wuxi, Jiangsu, China

## ABSTRACT

Increasing research has shown that the abnormal expression of microRNA (miRNA) is associated with many complex diseases. However, biological experiments have many limitations in identifying the potential disease-miRNA associations. Therefore, we developed a computational model of Three-Layer Heterogeneous Network based on the Integration of CircRNA information for MiRNA-Disease Association prediction (TLHNICMDA). In the model, a disease-miRNA-circRNA heterogeneous network is built by known disease-miRNA associations, known miRNA-circRNA interactions, disease similarity, miRNA similarity, and circRNA similarity. Then, the potential disease-miRNA associations are identified by an update algorithm based on the global network. Finally, based on global and local leave-one-out cross validation (LOOCV), the values of AUCs in TLHNICMDA are 0.8795 and 0.7774. Moreover, the mean and standard deviation of AUC in 5-fold cross-validations is 0.8777 +/−0.0010. Especially, the two types of case studies illustrated the usefulness of TLHNICMDA in predicting disease-miRNA interactions.

## INTRODUCTION

MicroRNAs (miRNAs) are a crucial class of non-coding RNAs (ncRNAs), and the length of miRNAs is roughly 20 nucleotides (*Yu et al., 2022*). MiRNAs play an important role in repressing gene expression at the post-transcriptional level (*Planell-Saguer & Rodicio, 2011*). MiRNAs are the key players in cell differentiation, proliferation, survival, and other biological processes (*Ouellet et al., 2006*) by binding to target messenger RNAs (mRNAs) (*Ambros, 2004*). For example, *Pekarsky et al. (2006)* demonstrated that miR-29 and miR-181Tcl1 can modulate the expression of chronic lymphocytic leukemia (CLL) and may be candidates for inhibiting the overexpression of Tcl1 in CLLs. In recent years, an increasing number of studies have proved that miRNA is related to many diseases, such as breast cancer (*Zhu et al., 2014*), prostatic neoplasms (*Latronico, Catalucci & Condorelli, 2007*), lung cancer (*Chen et al., 2019*) and so on. In particular, it has been proved that miRNA has an essential role in the development, progression, and pathogenesis of various diseases

Corresponding author
Zhibin Jiang, jnuszmtjzb@163.com

(*Adibzadeh Sereshgi et al., 2019*; *Vahdat Lasemi et al., 2019*). For instance, in the blood of patients with Alzheimer's disease (AD), 36 miRNA abnormalities were discovered by *Nagaraj et al. (2018)*, and it was established that those 36 miRNAs could be employed as potential biomarkers for the diagnosis of AD. Moreover, because of their important role in different diseases, miRNA molecules can be taken as potential therapeutic targets, therapeutic agents, and diagnostic biomarkers (*Aghaee-Bakhtiari et al., 2016*). For instance, SPC3649, an antagonist of miR-122, can inhibit the replication of the hepatitis C virus in hepatocytes (*Pekarsky et al., 2006*). Therefore, identifying disease-associated miRNAs can provide a novel perspective in the realms of medical diagnosis, prevention, and treatment for complex human diseases (*Ambros, 2004*; *Bartel, 2004*). Currently, biological experiments are the traditional methods to identify the miRNA-disease association. However, these experimental methods are usually demanding and costly. Therefore, designing an effective calculating model for predicting the disease-miRNA associations can save time and money.

With the development of technology, many miRNA-disease associations were verified and published, and several miRNA-disease databases were constructed, such as HMDD v2.0 (*Li et al., 2014b*), miRCancer (*Xie et al., 2013*), HMDD v3.2 (*Huang et al., 2019*), and so on.

The assumption that functionally similar miRNAs may be linked with similar phenotypic diseases (*Aerts et al., 2006*) has prompted the creation of several calculating approaches for identifying possible disease-miRNA interactions. For example, *Chen, Liu & Yan (2012)* employed a Random Walk with Restart (RWR) algorithm to predict potential MiRNA-Disease Association (RWRMDA). In RWRMDA, RWR method was employed to calculate the association probability based on the constructed miRNA functional similarity network. However, RWRMDA is not work in a disease, which has any known related miRNA. *Chen et al. (2015)* employed a restricted Boltzmann machine (RBM) with multi-type miRNA-disease association to predict MiRNA-disease association (RBMMMDA). The RBM was employed to identify the potential miRNA-disease association with the different types based on the constructed miRNA-disease network. However, RBMMMDA does not apply to diseases where there is no known link between disease and miRNAs. *Shi et al. (2013)* mapped the disease genes and the miRNA genes into a protein-protein interaction (PPI) network. Based on the protein-protein interaction (PPI) network, the RWR method was employed to obtain the rank of gene similarity. However, this model is a local prediction model, and its prediction performance compared with the global prediction model is not excellent. *Mørk et al. (2014)* established a miRNA-protein-disease predicted model by linking miRNA with the disease through an association between miRNAs and related proteins. However, these methods do not show admirable prediction performance because their performance depends to a large extent on miRNA target interaction. *Jiang et al. (2010)* developed a miRNA similarity ranking algorithm with a cumulative hypergeometric distribution to infer miRNAs that may be associated with a disease. *Chen et al. (2016a)* predicted the potential miRNA-disease association by integrating the high-score miRNA (or disease) in known miRNA-disease associations and the high-score miRNA (or disease) in unknown miRNA-disease associations. In 2016,

*Chen et al. (2016b)* proposed a heterogeneous graph inference model to identify the MiRNA-disease association (HGIMDA) based on the constructed miRNA-disease network. HGIMDA created an interactive equation to identify the potential miRNA-disease association by summarizing all pathways with a length set to three in the constructed miRNA-disease network. However, experimental evidence for known miRNA-disease associations remains insufficient. A miRNA-disease prediction model TLHNMDA (*Chen et al., 2018*) was developed to predict the miRNA-disease association based on the constructed disease-miRNA-lncRNA heterogeneous network. In TLHNMDA, the potential miRNA-disease association was obtained by the information flow-based method in the constructed three-layer network. *Feng et al. (2023)* calculated the relabel neighbors of nodes in the constructed miRNA-disease network to reconstruct the miRNA-disease network for predicting the potential miRNA-disease associations.

Nowadays, machine learning has been widely used in various domains, and an increasing number of people have paid attention to it in bioinformatics. Machine learning also has many applications in disease miRNA-related prediction. *Jiang et al. (2013)* mined the potential association between miRNA and disease by a support vector machine (SVM) classifier. Based on the known disease-miRNA relationship, *Li et al. (2017)* developed a matrix completion method to predict the MiRNA-disease association (MCMDA). MCMDA constructed a disease-miRNA relationship matrix by the known disease-miRNA association and employed the singular value thresholding (SVT) method to obtain the potential disease-miRNA association. However, MCMDA relied on known disease-miRNA associations, which may have resulted in the low accuracy of the model. *Chen & Yan (2014)* employed the regularized least squares method to predict the potential miRNA-disease associations (RLSMDA). In RLSMDA, a semi-supervised classifier was designed to obtain the miRNA-disease association probability in miRNA space and the disease space, respectively. Finally, the final predicted score was obtained by combining the miRNA-disease association probability in the two different spaces. *Xuan et al. (2013)* developed a miRNA-disease prediction model by using weighted K-nearest neighbors. However, the model is a local ranking method and still has room for proving accuracy. *Chen, Wu & Yan (2017)* designed a prediction method for miRNA-disease interaction based on ranking *k* neighbors by combining miRNA and disease multi-source similarity and the proven miRNA-disease association. By using the SVM sorting model, reordering these *k* neighbors. Eventually, the ultimate ranking of all disease-miRNA interactions is obtained based on weighting the ranking results. However, the predicted results of the RKNNMDA method tend to be biased toward miRNAs related to more known related diseases. *Zhang, Wei & Liu (2022)* proposed a ranking framework to predict miRNA-disease association based on a constructed miRNA-disease network. On the constructed heterogeneous network, the node2vec method (*Grover & Leskovec, 2016*) was employed to extract the features for both miRNA and disease, respectively. They employed five machine learning methods to calculate the scores of the miRNA-disease associations, respectively. Finally, the miRNA-disease association was obtained by using the LambdaMART method (*Burges, 2010*) to combine the potential probabilities in the five machine learning methods. *Huang et al. (2021)* proposed a tensor decomposition with relational constraints (TDRC)

model to identify the potential disease-miRNA association based on a 3D miRNA-disease-type tensor. This approach involved combining a 3D miRNA-disease-type tensor with known miRNA-disease associations, miRNA functional similarity, and disease semantic similarity matrices. *Yu, Zheng & Gao (2022)* constructed a miRNA-disease-gene heterogeneous network and employed meta-paths to predict potential miRNA-disease association. Defining seven types of meta-paths within the constructed three-layer network, they extracted features from each path type and integrated them to calculate the association probability between miRNA and disease. In the constructed miRNA-disease-gene network, *He et al. (2023a)* employed multiple graph convolutional networks with the Chebyshev filter as the encoder to obtain the embedding of nodes. Then, a linear decoder was employed to predict the potential miRNA-disease associations based on the embeddings of nodes. For reconstructing the miRNA-disease association network, *He et al. (2023b)* identified the functional module by constructing five types of higher-order Markov chains. Then, a path-based method was used to predict the potential miRNA-disease associations based on a reconstructed miRNA-disease network.

Recent research has revealed that circular RNA (circRNA) exhibits a distinct behavior compared to traditional linear RNAs. The closed-loop structure of circRNA, unaffected by exonucleases, leads to a more stable expression profile and reduced susceptibility to regression (*Chen & Yang, 2015*). Consequently, circRNA is recognized as a novel ncRNA. Some circRNAs act as natural miRNA sponges, regulating miRNA activity and modulating miRNA targets to alleviate their passive impact on hereditary factors (*Militello et al., 2016*). CircRNAs significantly contribute to the pathogenesis of various disorders, including atherosclerotic nervous system disorders, diabetes, and neoplasms, among others. A deeper understanding of circRNA structure and function has the potential to enhance our knowledge of pathogenic mechanisms, thereby improving disease prevention and diagnosis (*Han, Chao & Yao, 2018*; *Tang et al., 2021*; *Tang et al., 2020*). Several researches have been developed to predict the potential circRNA-disease association. For example, *Fu et al. (2023)* proposed a graph embedding method to predict the potential circRNA-disease association based on a constructed circRNA-miRNA-disease heterogeneous network (HGECDA). Using meta-path-based random walks in HGECDA, they captured interactions among circRNA, miRNA, and disease nodes. The path embedding model was then employed to obtain the embeddings of the node, and the CosMulformer model predicted the association between circRNA and disease. Additionally, in TLHNMDA, the lncRNA-miRNA associations were introduced to construct a disease-miRNA-lncRNA heterogeneous network for predicting the potential miRNA-disease associations. Especially, the lncRNA information played a role in connecting the miRNA and disease. Thus, building on the notion that circRNA is associated with disease through miRNA and inspired by TLHNMDA, we incorporated circRNA-miRNA associations to construct a disease-miRNA-circRNA heterogeneous network for predicting the potential miRNA-disease associations.

In this research, we developed a Three-Layer Heterogeneous Network model with the Integrated CircRNA information to identify the potential MiRNA-disease association

(TLHNICMDA) by integrating multi-source data, which is theoretically feasible. Firstly, a disease-miRNA heterogeneous network was created by multi-type similarity data of diseases and miRNAs. At the same time, ground on the proven miRNA-circRNA relationships, the Gaussian interaction profile kernel similarity network of circRNA was combined with the constructed disease-miRNA heterogeneous network to form a disease-miRNA-circRNA heterogeneous network. Finally, TLHNICMDA introduced an updated algorithm to infer the interaction between diseases and miRNAs. TLHNICMDA can fully mine the topological information of the three-layer heterogeneous network and effectively recognize the disease-miRNA associations. Be confronted with the local area network prediction, the prediction accuracy of TLHNICMDA was improved. At the same time, the model can also identify the association between circRNA and miRNA. To assess the effectiveness of TLHNICMDA, three types of cross-validation methods were employed. Especially, the AUC values of the three validation methods are 0.8795, 0.7774, and 0.8777 +/−0.0010, respectively. Four complex human diseases were selected as case studies, and out of the top 50 miRNAs identified in kidney tumors, pancreatic tumors, breast tumors, and lung tumors, 40, 40, 45, and 41 have been validated by at least one database among miR2Disease, dbDEMC, and HDMM v2.0. These outcomes indicate that TLHNICMDA has an excellent performance in predicting the interaction between disease and miRNA.

## MATERIALS AND METHODS

### Human miRNA-disease associations

We collected the human miRNA-disease associations from the HMDD v2.0 database (http://www.cuilab.cn/hmdd) (*Li et al., 2014b*). The HMDD v2.0 database contains 5,430 known miRNA-disease associations between 495 miRNAs and 383 diseases. Here, we defined an adjacency matrix $A^m \in R^{nd \times nm}$ to store the known disease-miRNA associations, where $nd$ is the number of diseases and $nm$ is the number of miRNAs. If there is a known association between miRNA $j$ and disease $i$, then $A^m(i,j) = 1$, otherwise $A^m(i,j) = 0$. Therefore, the adjacency matrix is referred to Eq. (1).

$$A^m(i,j) = \begin{cases} 1, & \textit{If miRNA j and disease i have a known interaction} \\ 0, & \textit{Otherwise} \end{cases} \quad (1)$$

### Human circRNA-miRNA interaction

In 2011, *Yang et al. (2011)* developed the starBase database to facilitate comprehensive exploration of miRNA-targets interactions. As of May 21, 2023, the starBase v2.0 database (http://starbase.sysu.edu.cn/starbase2/index.php) (*Li et al., 2014a*) has collected approximately 1,600,000 pairs of miRNA-ncRNA interactions, from which we selected circRNAs associated with 495 miRNAs. Finally, we collected 58,979 known circRNA-miRNA interaction pairs involving 495 miRNAs and 6,390 circRNAs from the starBasev2.0. Subsequently, an association matrix $B^m \in R^{nm \times nc}$ was built to store these known circRNA-miRNA interaction pairs, where $nc$ indicates the number of circRNAs. If

the relationship between miRNA $i$ and circRNA $j$ is unsupported, then $B^m(i,j) = 0$; otherwise, it is set to 1. Thus, the matrix $B^m$ is referred to Eq. (2).

$$B^m(i,j) = \begin{cases} 1, & \textit{If circRNA j and miRNA i have a known interaction} \\ 0, & \textit{Otherwise} \end{cases} \tag{2}$$

## Disease semantic similarity model 1

As we all know, the structure of a directed acyclic graph (DAG) can be instrumental in deducing the relationships among diseases (*Cui, 2010*). Originating from MeSH (Medical Subject Heading; http://www.nlm.nih.gov/) descriptor of Category C, the disease DAG is shaped by both disease semantic similarity and its inherent structure. The edges within the DAG articulate a hierarchy, with each node representing a transition from a more general term (parent node) to a more specific term (child node) (*Wright et al., 2009*). For example, the DAG of disease $A$ can be expressed as $DAG_A = (A, T_A, E_A)$, where $T_A$ contains the set of node $A$ and the ancestral node of $A$, and $E_A$ is the edge set of disease $A$. For disease $t$ in $DAG_A$, the semantic contribution score of the disease is referred to Eq. (3):

$$\begin{cases} D_A 1(t) = 1, & \textit{If } t = A \\ D_A 2(t) = \max\{\Delta * D_A(t') \in \textit{children } t\}, & \textit{If } t \neq A \end{cases} \tag{3}$$

The semantic score of disease $A$ to itself in the DAG is specified as 1, reflecting that disease $A$ is the most particular disease by itself. Especially, as the distance between nodes in the DAG increases, the value of the ancestor node's contribution to the semantic score of the offspring node diminishes. To model this decay in contribution, $\Delta \in (0,1)$ is introduced as the decay factor, mitigating the influence of distant ancestor nodes on disease $A$. Following the previous study, $\Delta$ is specified as 0.5 (*Ping et al., 2013*). The semantic value $DV1(A)$ of disease $A$ in $DAG_A$ is calculated as Eq. (4):

$$DV1(A) = \sum_{t \in T_A} D_A 1(t) \tag{4}$$

As shown in Fig. 1, the DGA diagram of Bacteremia, the semantic value of Bacteremia = 1 (Bacteremia) + 0.5 (Bacterial Infections) + 0.25 (Bacterial Infections and Mycoses) + 0.5 (Sepsis) + 0.2 (Infection) + 0.25 (Systemic Inflammatory Response Syndrome) + 0.125 (Inflammation) + 0.0625 (Pathologic Processes) + 0.03125 (Pathological Conditions, Signs and Symptoms) = 2.46875. The assumption suggested that the higher the value of the semantic similarity between diseases, the greater similar they are in the DAG (*Wang et al., 2021*). The semantic similarity between disease $B$ and disease $A$ in semantic similarity model 1 is described as Eq. (5):

$$SS1(A,B) = \frac{\sum_{t \in T_A \cap T_B} (D_A 1(t) + D_B 1(t))}{DV1(A) + DV1(B)} \tag{5}$$

where $D_A 1(t)$ and $D_B 1(t)$ are the semantic values of disease $t$ for disease $A$ and disease $B$, separately. $DV1(\cdot)$ is the semantic score of disease. $SS1$ represents the first semantic similarity matrix of disease.

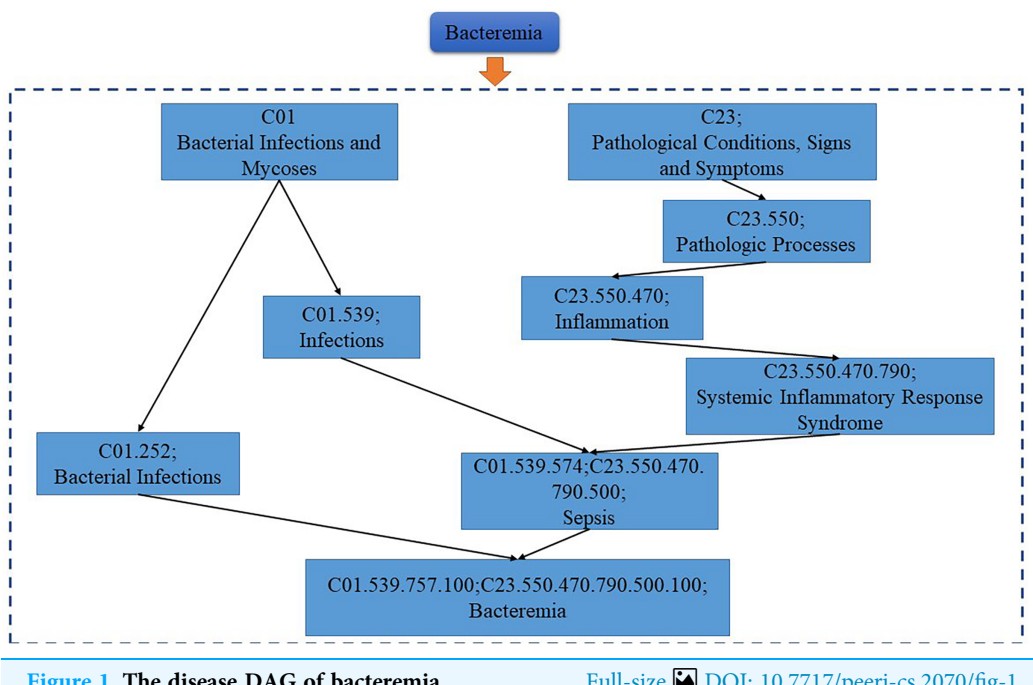

**Figure 1 The disease DAG of bacteremia.** 

## Disease semantic similarity model 2

Considering the same contribution score of disease $t$ to $A$ in the same layer of DAG(A) is inaccurate, a second disease semantic similarity model 2 was proposed (*van Laarhoven, Nabuurs & Marchiori, 2011*; *Ping et al., 2013*). In the disease semantic similarity model 2, the semantic contribution of disease $t$ to disease $A$ is obtained by Eq. (7).

$$D_A2(t) = -\log\left[\frac{Number\ of\ DAGs\ including\ t}{Number\ of\ disease}\right] \tag{6}$$

$$DV2(A) = \sum_{t\in T_A} D_A2(t) \tag{7}$$

In the disease semantic similarity calculation model 2, the semantic similarity between disease $A$ and disease $B$ is described as Eq. (8).

$$SS2(A,B) = \frac{\sum_{t\in T_A\cap T_B}(D_A2(t) + D_B2(t))}{DV2(A) + DV2(B)} \tag{8}$$

where $D_A2(t)$ and $D_B2(t)$ are the semantic values of disease $t$ for disease $A$ and disease $B$. $DV2(A)$ and $DV2(B)$ are the semantic scores of disease $A$ and disease $B$, separately. The SS2 represents the second disease similarity matrix.

## Gaussian interaction profile kernel similarity for diseases, miRNAs, and circRNAs

The radial basis function (RBF) was used in the Gaussian kernel function and measured the Euclidean distance between two samples, and the formula of RBF is defined as Eq. (9).

$$k(x, x') = \exp\left(-\left(\frac{||x - x'||}{2\sigma^2}\right)\right) \tag{9}$$

where $\sigma$ is the bandwidth parameter and $k(x, x')$ is the similarity between the vector $x$ and the vector $x'$.

The Gaussian interaction profile kernel similarities for diseases and miRNAs were obtained from proven disease-miRNA association. The Gaussian interaction profile kernel similarity of circRNAs was calculated based on the known circRNA-miRNA association. Additionally, the Gaussian interaction profile kernel similarity is not sparse. We first defined a dualistic vector $IP(d_i)$ to represent i-th row in the matrix $A^m$ and $IP(m_i)$ denotes the i-th column in the matrix $A^m$. $IP(c_i)$ is the i-th in the matrix $B^m$. Therefore, the Gaussian interaction profile kernel similarity for disease $d_i$ and disease $d_j$ is obtained by the Eq. (10) (*Chen et al., 2017*; *van Laarhoven, Nabuurs & Marchiori, 2011*). Similarly, the Gaussian interaction profile kernel similarity for miRNA $m_i$ and miRNA $m_j$ is calculated by the Eq. (11), the Gaussian interaction profile kernel similarity of circRNA $c_i$ and circRNA $c_j$ is described in Eq. (12).

$$KD(d_i, d_j) = \exp(-\gamma_d \, ||IP(d_i) - IP(d_j)||^2) \tag{10}$$

$$KM(m_i, m_j) = \exp(-\gamma_m \, ||IP(m_i) - IP(m_j)||^2) \tag{11}$$

$$KC(c_i, c_j) = \exp(-\gamma_c \, ||IP(c_i) - IP(c_j)||^2) \tag{12}$$

where $KD$ is the Gaussian interaction profile kernel similarity for disease, $KM$ is the Gaussian interaction profile kernel similarity for miRNA, $KC$ is the Gaussian interaction profile kernel similarity of circRNA, $r_d$, $\gamma_m$ and $\gamma_c$ are the normalized Gaussian kernel bandwidth and defined as follows:

$$\gamma_d = \frac{\gamma_d{}'}{\left(\frac{1}{nd}\sum_{i=1}^{nd} ||IP(d_i)||^2\right)} \tag{13}$$

$$\gamma_m = \frac{\gamma_m{}'}{\left(\frac{1}{nm}\sum_{i=1}^{nm} ||IP(m_i)||^2\right)} \tag{14}$$

$$\gamma_c = \frac{\gamma_c{}'}{\left(\frac{1}{nc}\sum_{i=1}^{nc} ||IP(c_i)||^2\right)} \tag{15}$$

where $\gamma_d{}'$, $\gamma_m{}'$ and $\gamma_c{}'$ are the original width, which are set to 1 from the previous study.

### Integrated similarity for diseases

To reduce the sparse of the disease semantic similarity, the integrated similarity was used. Additionally, the disease semantic similarity was obtained by averaging the value of the disease semantic similarity 1 and the disease semantic similarity 2. The integrated similarity network of disease was constructed by Gaussian interaction profile

kernel similarity and the disease semantic similarity, and the formula can be described as Eq. (16):

$$SD(d_i, d_j) = \begin{cases} \dfrac{SS1(d_i, d_j) + SS2(d_i, d_j)}{2}, & \text{If disease } d_i \text{ and } d_j \text{ have semantic similarity} \\ KD(d_i, d_j), & \text{Otherwise} \end{cases} \quad (16)$$

where $SD$ is the constructed disease similarity matrix, $SS1$ denotes the disease semantic similarity 1, and $SS2$ is the disease semantic similarity 2.

## MiRNAs functional similarity

Based on the conception that functionally similar miRNAs tend to interact with similar diseases, *Cui (2010)* calculated the semantic similarity between diseases associated with miRNA to obtain the functional similarity of miRNA. The calculated process of functional similarity for miRNA is divided into four steps. First, we constructed the disease set $D(m_i)$ that the disease is associated with miRNA $m_i$, and the disease dataset $D(m_j)$ that disease is related to miRNA $m_j$. Second, based on the constructed disease datasets, the disease semantic contribution value is calculated. Third, the disease semantic similarity is computed. Finally, the functional similarity for miRNA is calculated by the disease semantic similarity in step three. The functional similarity of miRNA was from 'misim.zip' at http://www.cuilab.cn/files/images/cuilab. $FS$ is the functional similarity of miRNA.

## Integrated similarity for MiRNAs

Similarly, the integrated miRNA similarity was used to reduce the sparse of the miRNA functional similarity. The integrated miRNA similarity network was built by the functional similarity of the miRNAs and the Gaussian interaction profile kernel similarity of miRNAs, and the formula of the integrated miRNA similarity is described as Eq. (17):

$$SM(m_i, m_j) = \begin{cases} FS(m_i, m_j), & \text{if } m_i \text{ and } m_j \text{ have functional similarity} \\ KM(m_i, m_j), & \text{Otherwise} \end{cases} \quad (17)$$

where $SM$ is the constructed integrated similarity network of miRNAs. Especially, we have normalized the above similarities.

## TLHNICMDA

The TLHNICMDA model proposed to enhance the predictive precision by incorporating additional data types to compensate for the low accuracy of the model because the training data is lacking. Various interactions, such as miRNA-mRNA interaction, miRNA-environmental factor interaction, miRNA-lncRNA, and miRNA-circRNA interaction, can be introduced into the TLHNICMDA model. However, integrating diverse data types may also bring about additional noise information. In particular, circRNA is associated with many diseases by modifying the miRNA. Therefore, we introduced the circRNA-miRNA association to build a comprehensive and reliable association network. Subsequently, we developed a potent and reasonable calculation model to infer possible miRNA-disease

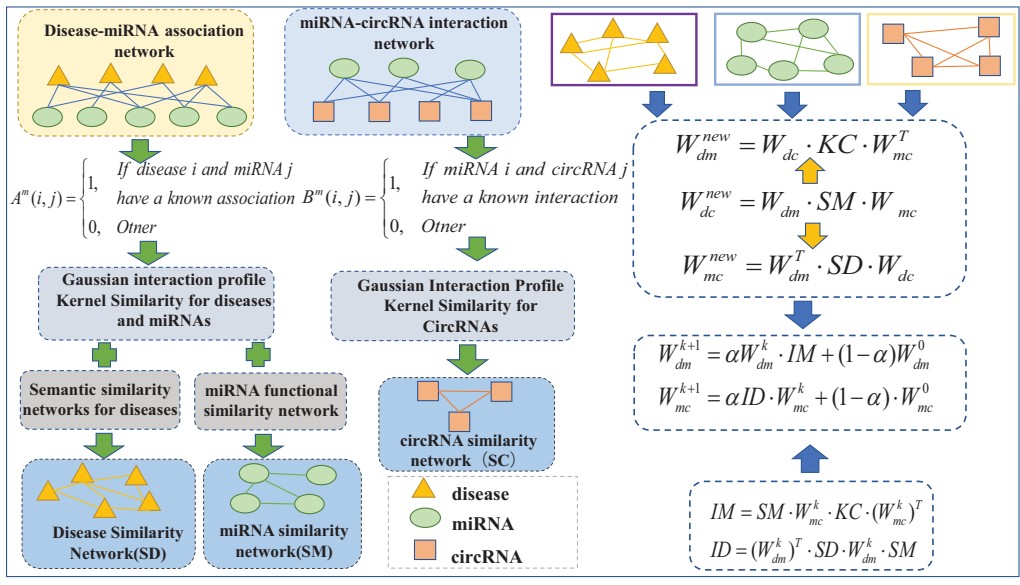

**Figure 2 The model flow chart of TLHNICMDA.**

interactions. TLHNICMDA can also identify possible circRNA-miRNA interactions. The detail of TLHNICMDA is shown in Fig. 2.

In this study, we constructed a three-layer heterogeneous network with disease, miRNA, and circRNA as the three types of nodes. Edges between identical nodes represented their similarity, while edges between different nodes indicated associations or interactions. Since the association between disease $d$ and circRNA $c$ was unknown, $W_{dc}$ was calculated based on Fig. 3C pathways. For disease $d$ and circRNA $c$, ground on the interaction between miRNA $m$ and disease $d$, the interaction between miRNA $m$ and circRNA $c$, and miRNA similarity, all pathways of length were set to three between circRNA $c$ and disease $d$ were searched pathways, and the interaction value $W_{dc}$ between circRNA $c$ and disease $d$ was obtained according to Eq. (18). For miRNA $m$ and disease $d$, ground on the miRNA-circRNA interaction data, disease-miRNA interactions and circRNA similarity, all paths of length were set to three between miRNA $m$ and disease $d$ were searched as well as the interaction probability $W_{dm}$ can be obtained according to Eq. (19), as shown in Fig. 3A. For miRNA $m$ and circRNA $c$, based on disease-miRNA association, circRNA similarity as well as disease-circRNA association, all pathways of length were set to three between miRNA $m$ and circRNA $c$ can be searched as well as the relation probability $W_{mc}$ between circRNA $c$ and miRNA $m$ can be calculated according to Eq. (20), as shown in Fig. 3B.

$$W_{dc}(d, c) = \sum_{m_i \in M} \sum_{m_j \in M} W_{dm}(d, m_i) \cdot SM(m_i, m_j) \cdot W_{mc}(m_j, c) \qquad (18)$$

$$W_{dm}(d, m) = \sum_{c_i \in C} \sum_{c_j \in C} W_{dc}(d, c_i) \cdot KC(c_i, c_j) \cdot W_{mc}^T(m, c_j) \qquad (19)$$
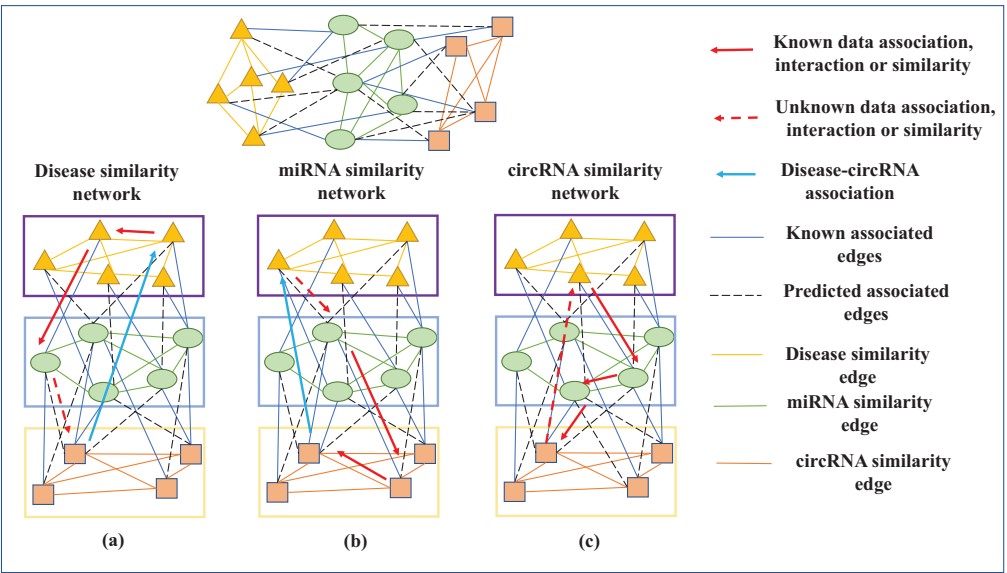

**Figure 3 The calculation process of disease-miRNA association score W$_{dm}$, miRNA-circRNA interaction score W$_{mc}$ and disease-circRNA association score W$_{dc}$.**

$$W_{mc}(m, c) = \sum_{d_i \in D} \sum_{d_j \in D} W_{dm}^T(d_i, m) \cdot SD(d_i, d_j) \cdot W_{dc}(d_j, c) \qquad (20)$$

In the above equation, $W_{dm}(d, m_i)$ is the interaction probability between miRNA $m_i$ and disease $d$, while $W_{mc}(m, c_j)$ is the interaction probability between miRNA $m$ and circRNA $c_j$, $W_{dc}(d_j, c)$ is the relation probability between circRNA $c$ and disease $d_j$.

Three matrixes were formed based on the above three formulas defined as follows:

$$W_{dm}^{new} = W_{dc} \cdot KC \cdot W_{mc}^T \qquad (21)$$
$$W_{dc}^{new} = W_{dm} \cdot SM \cdot W_{mc} \qquad (22)$$
$$W_{mc}^{new} = W_{dm}^T \cdot SD \cdot W_{dc} \qquad (23)$$

where $W_{dm}$ is the disease-miRNA interaction probability matrix, $W_{mc}$ is the interaction fraction matrix between circRNA and miRNA, $W_{dc}$ is the association fraction matrix between disease and circRNA. The matrix superscript $T$ refers to the transpose of the corresponding matrix. Using Eq. (22) below with equation as an intermediate value instead of $W_{dc}$ is because the data of $W_{dc}$ cannot be expressed directly, so $W_{dm}$ and $W_{mc}$ can be written as the following Eqs. (24) and (25):

$$W_{dm}^{new} = W_{dm} \cdot SM \cdot W_{mc} \cdot KC \cdot W_{mc}^T \qquad (24)$$
$$W_{mc}^{new} = W_{dm}^T \cdot SD \cdot W_{dm} \cdot SM \cdot W_{mc} \qquad (25)$$

Since the initially known disease-miRNA association and known circRNA-miRNA interaction data have been verified by previous experiments to be credible. Therefore, it

can be added to Eqs. (26) and (27) with certain weights, finally the global iteration formula is constructed as follows:

$$W_{dm}^{k+1} = \alpha W_{dm}^k \cdot (SM \cdot W_{mc}^k \cdot KC \cdot (W_{mc}^k)^T) + (1 - \alpha) W_{dm}^0 \tag{26}$$

$$W_{mc}^{k+1} = \alpha((W_{dm}^k)^T \cdot SD \cdot W_{dm}^k \cdot SM) \cdot W_{mc}^k + (1 - \alpha) \cdot W_{mc}^0 \tag{27}$$

where $\alpha$ is the regulatory weight between 0 and 1, $W_{dm}^0$ and $W_{mc}^0$ are the original interaction matrix of proven miRNA-disease and the interaction matrix of circRNA-miRNA, respectively, $W_{dm}^k$ is described as the potential disease-miRNA association in the k-th iteration, $W_{mc}^k$ denotes the predicted miRNA-circRNA association in the k-th iteration. To simplify Eqs. (26) and (27), we introduced two variables IM and ID. The formula of IM can be described as Eq. (28) and the formula of ID is defined as Eq. (29):

$$IM = SM \cdot W_{mc}^k \cdot KC \cdot (W_{mc}^k)^T \tag{28}$$

$$ID = (W_{dm}^k)^T \cdot SD \cdot W_{dm}^k \cdot SM \tag{29}$$

Since then, the Eqs. (26) and (27) can be transformed into Eqs. (30) and (31), respectively.

$$W_{dm}^{k+1} = \alpha W_{dm}^k \cdot IM + (1 - \alpha) W_{dm}^0 \tag{30}$$

$$W_{mc}^{k+1} = \alpha ID \cdot W_{mc}^k + (1 - \alpha) \cdot W_{mc}^0 \tag{31}$$

Especially, to ensure the solution of Eqs. (30) and (31) converge (the proof was provided in the Supplemental Materials), IM and ID are normalized by Eqs. (32) and (33), respectively.

$$IM(m(i), m(j)) = \frac{IM(m(i), m(j))}{\sqrt{\sum_{c=1}^{nm} IM(m(i), m(c))} \cdot \sqrt{\sum_{c=1}^{nm} IM(m(j), m(c))}} \tag{32}$$

$$ID(d(i), d(j)) = \frac{ID(d(i), d(j))}{\sqrt{\sum_{c=1}^{nd} ID(d(i), d(c))} \cdot \sqrt{\sum_{c=1}^{nd} ID(d(j), d(c))}} \tag{33}$$

When the distance in the L1 norm between $W_{dm}^{k+1}$ and $W_{dm}^k$ is less than $10^{-6}$, the iteration is stopped. The final converging values $W_{dm}^{k+1}$ and $W_{mc}^{k+1}$ are the possible disease-miRNA interaction predicted score matrix and the potential predicted score matrix between circRNA and miRNA, respectively.

In the TLHNICMDA, the semantic similarity model 1, the semantic similarity model 2, and the Gaussian interaction profile kernel similarity network of diseases were integrated into the disease similarity network. Similarly, the miRNA functional similarity network was combined with the miRNA functional similarity and the miRNA Gaussian interaction profile kernel similarity. The circRNA similarity network was built by the proven circRNA-miRNA association and the Gaussian interaction profile kernel similarity of circRNA. On this basis, a three-layer heterogeneous network was built to improve the conundrum of low model prediction accuracy as a result of a lack the training numbers. We proposed an iterative update algorithm to identify possible disease-miRNA

interactions. In each iteration, the miRNA-disease interaction score was calculated to update the circRNA-miRNA association and disease-miRNA interaction in the next iteration. At the same time, the interaction score between miRNA and circRNA was calculated in each iteration to update the disease-miRNA association and circRNA-miRNA interaction in the next iteration.

## RESULTS

### Performance evaluation

In this research, the 5,430 pairs of known disease-miRNA relationships about 495 miRNAs and 383 diseases from the database HMDD v2.0 were used as training data, and 5-fold cross-validation, global LOOCV, and local LOOCV were used to evaluate TLHNICMDA. In global LOOCV, a test sample was selected one by one from the known disease-miRNA interactions as well, the training samples were the remaining 5,429 known disease-miRNA associations, and all the unproved disease-miRNA associations were taken as examinee samples. Training samples were employed to train the model. Then the model obtained after training was tested using the test sample. Finally, the values of all the test samples were obtained. In the global LOOCV, we combined the scores of the test samples and the candidate samples, and the rank of the test samples was obtained by ranking them. Different from the global LOOCV, the candidate samples in local LOOCV consisted of miRNAs that were not related to the disease in the test sample. In local LOOCV, the ranking of test sample was determined by sorting the scores of each test sample within the candidate samples, and finally got a ranking of 5,430 test samples. For five-fold cross-validation, all proven disease-miRNA connections were divided randomly into five equal-sized sets, with the surplus four subsets serving as training samples. Each of the five subgroups served as a test sample in turn. Equally, the candidate samples were unproved disease-miRNA pairs in five-fold cross-validation. The results of all candidate score samples were then compared to the scores of each sample for testing. Finally, the rank of all test sample scores was obtained. After five-fold cross-validation, we got the scores of 5,430 samples. In order to obtain a reliable performance evaluation, the process of five-fold cross-validation was repeatedly calculated 100 times. Finally, the predetermined threshold was employed to compare with the test sample scores rank. Especially, the test sample was categorized as an active sample if its score ranking exceeds the cutoff, meaning that the miRNA it contains is linked to the disease. The test sample will be labeled as a negative sample, meaning that the miRNA in the test sample is unrelated to the disease if the predetermined threshold score exceeds the test sample rank. In short, a higher miRNA-disease association score indicated a greater likelihood of interaction and *vice versa*. To evaluate the predicted performance of TLHNICMDA, the receiver operating characteristic (ROC) curve and the area under the ROC curve (AUC) were calculated. The abscissa of ROC curve is false positive rate (FPR), and the ordinate of ROC curve is true positive rate (TPR). In this study, we calculated the AUCs of TLHNICMDA and other three disease-miRNA association predicted methods in five-fold cross-validation, global LOOCV, and local LOOCV. Then compare the AUCs of TLHNICMDA with the other three models (Detailed comparison data are displayed in Table 1). As shown in Figs. 4 and 5, the other

**Table 1** The AUC value of global cross-validation, local cross-validation and five-fold cross-validation for the four model.

| Model name | AUC for global LOOCV | AUC for local LOOCV | AUC for 5-fold cross validation |
| --- | --- | --- | --- |
| TLHNICMDA | 0.8795 | 0.7774 | 0.8777+/−0.0010 |
| TLHNMDA | 0.8795 | 0.7756 | 0.8795+/−0.0010 |
| MCMDA | 0.8748 | 0.7606 | 0.8757+/−0.0011 |
| RLSMDA | 0.8426 | 0.6953 | 0.8569+/−0.0020 |

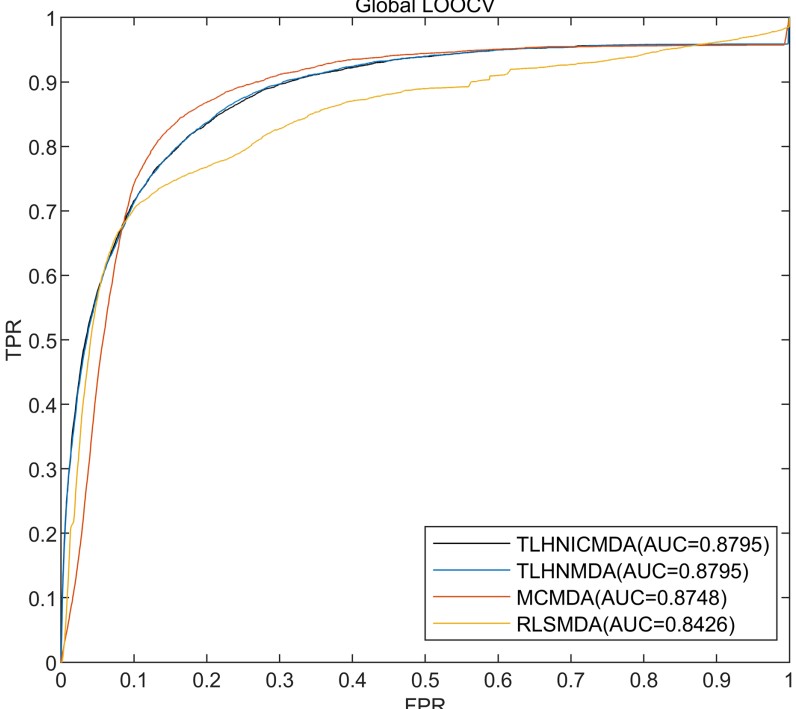

**Figure 4** The AUC value of global LOOCV for the four models.

three prediction models are TLHNMDA, MCMDA, and RLSMDA. In the global LOOCV AUC values for TLHNICMDA, TLHNMDA, MCMDA, and RLSMDA are 0.8795, 0.8795, 0.8748, and 0.8426, respectively. In local LOOCV. Their respective AUC values are 0.7774, 0.7756, 0.7606, and 0.6953, separately. In global LOOCV, the AUC comparison results show that TLHNICMDA and TLHNMDA demonstrate outstanding predictive performance. However, the AUC of TLHNICMDA in local LOOCV is the highest among the four models. In the 5-fold cross-validation, the AUC calculated by the above model is 0.8777+/−0.0010, 0.8795+/−0.0010, 0.8757+/−0.0011, and 0.8569+/−0.0020, separately. In 5-fold cross-validation, TLHNICMDA was the highest AUC value except for TLHNMDA. As shown in Table 1, TLHMICMDA outperformed the model MCMDA, and RLSMDA in global LOOCV, local LOOCV, and 5-fold cross-validation. Compared with TLHNMDA, TLHNICMDA outperformed the model TLHNMDA in local LOOCV, and the AUC

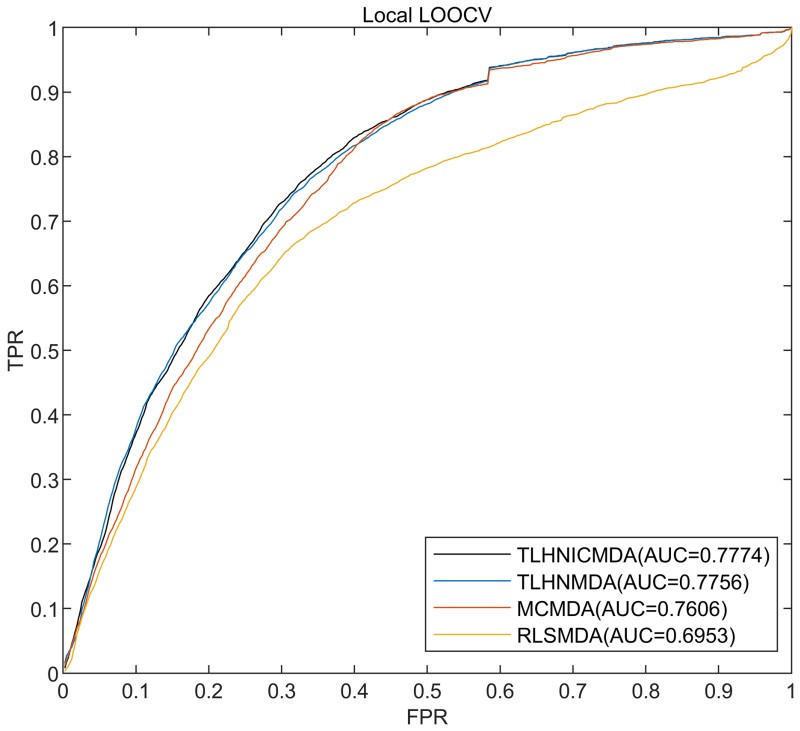

**Figure 5 The AUC value of local LOOCV for the four models.**

values of TLHNICMDA and TLHNMDA were equal in global LOOCV. In particular, we used the new disease-miRNA associations collected from the HMDD v3.2 database (*Zhou et al., 2018*) to evaluate the fitness of TLHNICMDA in a new database for 5-fold cross-validation, global LOOCV, and local LOOCV. Especially, the new disease-miRNA interactions were 8,968 confirmed miRNA-disease association pairs between 788 miRNAs and 374 diseases collected from the HMDD v3.2 database. Additionally, the characteristic data for miRNAs and diseases was gathered, which includes the miRNA functional similarity, the disease semantic similarity, miRNA sequence similarity, miRNA semantic similarity, and the Gaussian interaction profile kernel similarity of miRNA and disease (*Ding et al., 2022*). In global LOOCV, TLHNICMDA achieved an AUC of 0.9056 and in local LOOCV, it attained an AUC of 0.8299. In 5-fold cross-validation, TLHNICMDA maintained an AUC of 0.9253 +/−0.0010.

From the previous study (*Davis & Goadrich, 2006*; *He & Garcia, 2009*), the Precision-Recall (PR) curve can evaluate the performance of model in imbalanced datasets and provide more information than ROC curve. PR curve is plotted by the relationship between precision and recall. In our study, the known miRNA-disease associations were far less than the unknown miRNA-disease associations. Thus, we added the PR curve to our study for further evaluating the performance of models. The PR curve is plotted by using the rank result of global LOOCV in Fig. 6. In Fig. 6, TLHNICMDA outperformed the model TLHNMDA, MCMDA, and RLSMDA. Overall, TLHNICMDA can effectively predict the potential miRNA-disease association in the imbalanced dataset.

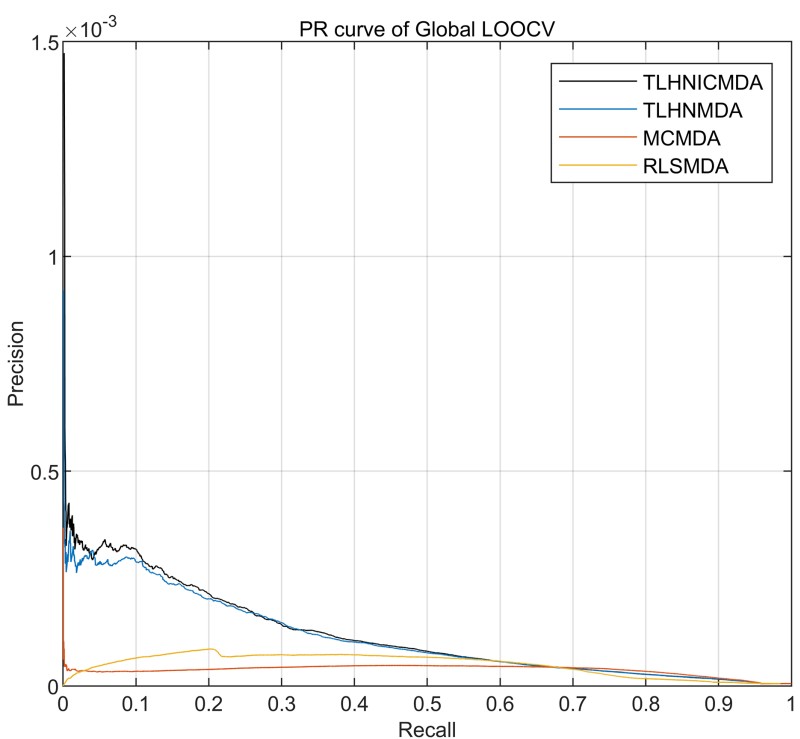

**Figure 6 Performance comparisons between TLHNICMDA and baseline models based on global LOOCV.**

We further applied the paired t-tests to analyze the significance between TLHNICMDA and the compared models (TLHNMDA, MCMDA, RLSMDA) by using the rank result in global LOOCV and local LOOCV based on the HMDD v2.0 database, respectively. In the global LOOCV, the $P$-values between TLHNICMDA and other models (TLHNMDA, MCMDA, RLSMDA) were 0.877, 1.4436E−04, and 7.2632E−26. In the local LOOCV, the $P$-values between TLHNICMDA and other models (TLHNMDA, MCMDA, RLSMDA) were 0.2009, 0.0443, and 5.6950E−92. In the HMDD v2.0, the compared results of the $P$-value suggested that TLHNICMDA is significantly different from the three models (TLHNMDA, MCMDA, RLSMDA).

## Time complex analysis

As shown in Fig. 2, there are mainly two steps for predicting the potential miRNA-disease association, which are the constructed similarity networks and the network iterations. Therefore, we analyzed the time complexity for them one by one.

First, for $nd$ diseases, $nm$ miRNAs, and $nc$ circRNAs, TLHNICMDA calculated the disease semantic similarity1, the disease semantic similarity 2, the miRNA functional similarity, the Gaussian interaction profile kernel similarity of disease, miRNA, and circRNA, the integrated disease similarity and the integrated miRNA similarity. The time complexities of the above similarities are $O(nd^2)$, $O(nd^2)$, $O(nm^2)$, $O(nd^2)$, $O(nm^2)$, $O(nc^2)$, $O(nd^2)$, and $O(nm^2)$, respectively. We defined $m$ as the maximum number in disease, miRNA and circRNA. The time complexities of calculating similarities have all

changed as $O(m^2)$. The constructed similarity network in this step is $O(m^2) + O(m^2) + O(m^2)$. Thus, the total time complexity in the first step is $O(m^2) * 11 = O(m^2)$.

In the last step, the iteration of the model was stopped until the difference between the k-th iteration and the next iteration was less than $10^{-6}$. Especially, the time complexity of matrix multiplication between matrix $C \in R^{m \times n}$ and matrix $D \in R^{n \times m}$ is $O(m * n * m) = O(n * m^2)$. Following the above definition that $m$ is the maximum number in disease, miRNA, and circRNA, the time complexity of matrix multiplication is $O(m^3)$. In this step, we defined $x$ as the time of iteration. Thus, the total time complexity in this step is $O(x * m^3)$.

In summary, the total time complexity in TLHNICMDA is the sum in these two steps above, which is described as $O(m^2) + O(x * m^3)$.

From the above result, TLHNICMDA can identify the interactions between underlying diseases and miRNAs, effectively. Additionally, TLHNICMDA is a complex network method that is not complicated compared with machine learning methods and deep learning methods. In particular, the five-fold cross-validation was repeated 100 times and the run time was 11987.05425 s which was less than 4 h.

## Case studies

At present, we do not have the condition to do biological experiments for directly validating the predicted miRNA-disease associations. Thus, the case study was employed to validate the predicted miRNA-disease associations. In the case study, the predicted probabilities of miRNA-disease pairs were ranked, subsequently validating these predictions by cross-referencing with published literature. We employed training data sets from HMDD v1.0 and HMDD v2.0 to conduct case studies on several significant human complex diseases in order to further assess the TLHNICMDA's predicted accuracy. Two types of case studies and a total of four complex human diseases were studied. The first kind of case study used the 5,430 pairs of proven disease-miRNA association data from the HMDD v2.0 database as the training set to train TLHNICMDA. TLHNICMDA predicted miRNAs that may be associated with kidney tumors and colonic neoplasms, and based on the predicted scores ranks these miRNAs. The predicted score of the miRNA-disease pair denotes the association probability between miRNA and disease. Besides the 5,430 known miRNA-disease associations, the number of the predicted miRNA-disease associations between 383 diseases and 495 miRNAs is 184,155. In this study, we sorted the predicted probabilities of the miRNA-disease associations in descending order. The high predicted score denotes there is an association between miRNA and disease. Additionally, the predicted miRNA-disease association can further provide the direction for disease diagnosis, prevention, and treatment. The rear 50 and the middle 50 miRNA-disease associations in the whole rank of the predicted probabilities were low, which means they may not have an association. Additionally, the middle 50 of the predicted miRNA-disease associations were ranked in the range from 92,077 to 92,127. Thus, the top 50 of the rank were most valuable. In the future, we hope the researchers can verify those predicted miRNA-disease associations. Additionally, the top 50 predicted miRNA-disease

associations were most proven from the previous studies. Thus, the top 50 predicted results were selected to verify the performance of the model. Finally, the top 50 prediction results of renal kidney neoplasms and colonic neoplasms were then validated using two other significant disease-miRNA interaction databases, miR2Disease (*Jiang et al., 2009*) and dbDEMC (*Zhen et al., 2010*). For further evaluating the performance of TLHNICMDA in the new database, the HMDD v1.0 was used. HMDD v1.0 has 1,395 known miRNA-disease associations between 271 miRNAs and 137 diseases. The data in HMDD v1.0 was employed to train TLHNICMDA to predict breast and esophageal neoplasms related to miRNAs as another case study and sort these miRNAs based on the prediction score. Finally, two other disease-miRNA association databases, miR2Disease and dbDEMC, were combined with HMDD v2.0 to verify the first 50 prediction results of breast and esophageal neoplasms.

In the human urinary and male reproductive system, kidney neoplasms are one of the common neoplasms, of which 85% are renal cancer (*Damjanov & Mikuz, 2007*). With the development of tumor stem cell miRNA and high-throughput research techniques, new progress has been made in the experimental study of kidney neoplasms in various aspects. *Gottardo et al. (2007)* found that in the microarray of 161 miRNAs in human kidney tissues and adjacent normal tissues, let-7f-2, miR-27, miR-28, and miR-185 were highly expressed. Compared with normal kidney tissue and other types of tumors, the expression degrees of 5 miRNAs are generally elevated in nephroblastoma, which are miR-92, miR-19b, miR-18a, miR-20a and miR-17-5p (*Jia et al., 2013*). *Perdomo (2000)* found that a novel transcriptional suppressor ZnFN1A4 in the Ikaros family can be detected in normal kidney tissues. *Wang & Sun (2018)* found that the degree expression of miR-375 suppresses the diffusion of A-498 renal cell carcinoma by inducing apoptosis. Especially, the amplification of miR-375 can stop A-498 renal carcinoma cells from migrating and invading. *Liu et al. (2010)* also found that miR-23b is an oncogenic factor of kidney neoplasms, and now by shortening the expression of miR-23b, tumor growth can be inhibited, which is of considerable help to the treatment of this disease. *Senanayake et al. (2012)* found that miR-215, miR-200c, and miR-194 could be used as therapeutic targets for kidney neoplasms in children. In addition, more and more miRNAs have been found to link with the growth of kidney tumors. For predicting potential miRNAs associated with kidney tumors by TLHNICMDA, 5,430 known disease-miRNA interactions were taken as the data set and sorted miRNAs based on the obtained scores. Using dbDEMC and the miR2Disease database, the first 50 miRNAs were sustained to be potentially related to kidney cancers. Table 2 showed that among the top 10 miRNAs linked to kidney neoplasms only one miRNA was not validated by dbDEMC and miRDisease. In the first 50 miRNAs related to kidney neoplasms, 40 types of miRNAs were validated by at least one of the dbDEMC and miRDisease.

Colon tumor is one of the most frequent and lethal neoplasia (*Barbáchano et al., 2018*). As an RNA down-regulated in colon cancer 5-fluorouracil (5-FU)-resistant DLD-1 cells, miR-34a was verified (*Yukihiro et al., 2011*). Overexpressed miR-135b is linked with a bad prognosis for patients suffering from inflammatory bowel disease and sporadic colorectal cancer (*Valeri et al., 2013*). *Bauer & Hummon (2012)* identified a highly decreased

**Table 2 Verification of the top 50 miRNAs associated with kidney neoplasms predicted by the proposed model.**

| miRNA | Database | miRNA | Database |
|---|---|---|---|
| hsa-mir-103a | dbDEMC | hsa-mir-130b | dbDEMC |
| hsa-mir-107 | dbDEMC | hsa-mir-128 | Unconfirmed |
| hsa-mir-497 | dbDEMC | hsa-mir-873 | Unconfirmed |
| hsa-mir-15b | dbDEMC; miR2Disease | hsa-mir-202 | dbDEMC; miR2Disease |
| hsa-mir-424 | dbDEMC | hsa-mir-106b | dbDEMC |
| hsa-mir-195 | dbDEMC | hsa-mir-19a | dbDEMC; miR2Disease |
| hsa-mir-503 | dbDEMC; miR2Disease | hsa-mir-29c | dbDEMC; miR2Disease |
| hsa-mir-214 | Unconfirmed | hsa-mir-20a | dbDEMC; miR2Disease |
| hsa-let-7b | dbDEMC | hsa-mir-29a | dbDEMC |
| hsa-let-7c | dbDEMC | hsa-mir-130a | Unconfirmed |
| hsa-let-7i | dbDEMC | hsa-mir-301b | dbDEMC; miR2Disease |
| hsa-mir-98 | dbDEMC | hsa-mir-19b | Unconfirmed |
| hsa-let-7g | dbDEMC | hsa-mir-301a | dbDEMC |
| hsa-let-7a | dbDEMC | hsa-mir-18a | dbDEMC |
| hsa-let-7d | Unconfirmed | hsa-mir-18b | miR2Disease |
| hsa-let-7e | dbDEMC; miR2Disease | hsa-mir-17 | dbDEMC; miR2Disease |
| hsa-let-7f | dbDEMC | hsa-mir-20b | dbDEMC; miR2Disease |
| hsa-mir-27b | dbDEMC | hsa-mir-29b | dbDEMC; miR2Disease |
| hsa-mir-338 | dbDEMC; miR2Disease | hsa-mir-106a | dbDEMC |
| hsa-mir-27a | dbDEMC | hsa-mir-181a | Unconfirmed |
| hsa-mir-196a | dbDEMC | hsa-mir-96 | dbDEMC; miR2Disease |
| hsa-mir-196b | dbDEMC; miR2Disease | hsa-mir-182 | Unconfirmed |
| hsa-mir-7 | Unconfirmed | hsa-mir-4306 | dbDEMC |
| hsa-mir-454 | dbDEMC | hsa-mir-218 | dbDEMC |
| hsa-mir-93 | dbDEMC | hsa-mir-22 | Unconfirmed |

expression of the miRNA-143/145 cluster in multiple tumors like colorectal cancer, where both members exhibited antiproliferative properties through roles on cell cycle progression, invasion, or migration. *Dong et al. (2010)* and *He et al. (2012b)* found that in human colorectal cancer (CRC), miR-218 is down-regulated. The above research revealed that detecting the expression of miRNAs in colon tumor sufferers is crucial for analyzing the clinicopathological characteristics of colon neoplasm patients. TLHNICMDA was employed to identify the possible relationship between colon cancers and miRNAs. The predicted scores were the evidence to rank the potential miRNAs. Finally, Using the miR2Disease and dbDEMC database, the first 50 miRNAs were validated to be potentially linked with colon cancers. Table 3 showed that the database validated all the top 10 colon neoplasms with miRNA associations. Among the first 50 miRNAs related to colonic cancers, 40 types of miRNAs were validated by at least one of the miR2Disease and dbDEMC.

**Table 3 Verification of the top 50 miRNAs associated with colonic neoplasms predicted by the proposed model.**

| miRNA | Database | miRNA | Database |
|---|---|---|---|
| hsa-mir-103a | dbDEMC and miR2Disease | hsa-mir-454 | dbDEMC |
| hsa-mir-107 | dbDEMC and miR2Disease | hsa-mir-202 | Unconfirmed |
| hsa-mir-497 | miR2Disease | hsa-mir-873 | dbDEMC |
| hsa-mir-15b | dbDEMC | hsa-mir-128 | dbDEMC and miR2Disease |
| hsa-mir-15a | dbDEMC | hsa-mir-130b | dbDEMC and miR2Disease |
| hsa-mir-424 | dbDEMC and miR2Disease | hsa-mir-106b | dbDEMC |
| hsa-mir-195 | dbDEMC | hsa-mir-29c | dbDEMC and miR2Disease |
| hsa-mir-503 | dbDEMC | hsa-mir-19a | dbDEMC and miR2Disease |
| hsa-mir-214 | dbDEMC and miR2Disease | hsa-mir-20a | dbDEMC and miR2Disease |
| hsa-let-7b | dbDEMC | hsa-mir-29a | dbDEMC and miR2Disease |
| hsa-let-7c | dbDEMC | hsa-mir-19b | Unconfirmed |
| hsa-let-7i | Unconfirmed | hsa-mir-130a | Unconfirmed |
| hsa-mir-98 | dbDEMC and miR2Disease | hsa-mir-301b | dbDEMC and miR2Disease |
| hsa-let-7g | dbDEMC and miR2Disease | hsa-mir-18a | Unconfirmed |
| hsa-let-7a | dbDEMC | hsa-mir-301a | Unconfirmed |
| hsa-let-7d | dbDEMC | hsa-mir-18b | Unconfirmed |
| hsa-let-7e | dbDEMC and miR2Disease | hsa-mir-20b | dbDEMC and miR2Disease |
| hsa-let-7f | dbDEMC and miR2Disease | hsa-mir-29b | dbDEMC and miR2Disease |
| hsa-mir-27b | dbDEMC | hsa-mir-181a | dbDEMC and miR2Disease |
| hsa-mir-338 | miR2Disease | hsa-mir-96 | dbDEMC and miR2Disease |
| hsa-mir-27a | dbDEMC and miR2Disease | hsa-mir-182 | Unconfirmed |
| hsa-mir-196a | dbDEMC | hsa-mir-4306 | dbDEMC |
| hsa-mir-196b | dbDEMC and miR2Disease | hsa-mir-218 | dbDEMC |
| hsa-mir-7 | dbDEMC and miR2Disease | hsa-mir-22 | dbDEMC |
| hsa-mir-93 | Unconfirmed | hsa-mir-185 | Unconfirmed |

Breast cancer is a high clinical incidence unbenign tumor deriving from the ductal epithelium of the chest. Globally, breast cancer is still a problem for public health (*Veronesi et al., 2005*). In 2008, 1.38 million new breast cancer cases worldwide, accounted for 23% of the women's cancer cases (*Jemal et al., 2008*). At present, tumor markers such as genes and proteins (*Mirabelli & Incoronato, 2013*) have been increasingly widely used in preoperative diagnosis, guiding treatment and prognostic judgment of breast cancer. MiRNA acts as suppressors or carcinogens in a variety of carcinomas (*Dong et al., 2010*; *Feng et al., 2014*; *Tong et al., 2014*). *Mitra et al. (2011)* found that in MCF-7 cells with stable miRNA-mediated inhibition of JARID1B expression, regulatory changes of a variety of miRNAs were identified, including let-7E, a member of tumor suppressor miRNAs. Let-7 miRNA exists in a wide variety of species, and changes in expression levels of human let-7 miRNA family members have been linked with various types of carcinomas (*Jerome et al., 2007*). Let-7b may function as a cancer inhibitor marker in the progression of tumors in the breast by preventing the expression of breast cancer-causing cells, according to the discovery of

**Table 4 Verification of the top 50 miRNAs associated with breast neoplasms predicted by the proposed model.**

| miRNA | Database | miRNA | Database |
|---|---|---|---|
| hsa-mir-181d | dbDEMC; miR2Disease | hsa-mir-600 | dbDEMC |
| hsa-mir-448 | dbDEMC | hsa-mir-424 | dbDEMC |
| hsa-mir-186 | dbDEMC | hsa-mir-128a | miR2Disease |
| hsa-mir-124 | dbDEMC; HMDD | hsa-mir-362 | dbDEMC |
| hsa-mir-629 | dbDEMC; HMDD | hsa-mir-134 | dbDEMC |
| hsa-mir-377 | dbDEMC | hsa-mir-583 | dbDEMC |
| hsa-mir-769 | Unconfirmed | hsa-mir-501 | dbDEMC |
| hsa-mir-181a | dbDEMC; miR2Disease; HMDD | hsa-mir-95 | dbDEMC |
| hsa-mir-433 | dbDEMC | hsa-mir-520a | dbDEMC; HMDD |
| hsa-mir-154 | dbDEMC | hsa-mir-203 | dbDEMC; miR2Disease; HMDD |
| hsa-mir-602 | dbDEMC | hsa-mir-135a | dbDEMC; HMDD |
| hsa-mir-211 | dbDEMC | hsa-mir-208b | Unconfirmed |
| hsa-mir-539 | dbDEMC | hsa-mir-596 | Unconfirmed |
| hsa-mir-658 | dbDEMC | hsa-mir-376c | HMDD |
| hsa-mir-140 | dbDEMC; HMDD | hsa-mir-197 | dbDEMC; HMDD |
| hsa-mir-136 | dbDEMC; miR2Disease | hsa-mir-185 | dbDEMC |
| hsa-mir-431 | dbDEMC | hsa-mir-148a | dbDEMC; miR2Disease; HMDD |
| hsa-mir-421 | dbDEMC | hsa-mir-129 | dbDEMC; HMDD |
| hsa-mir-220 | Unconfirmed | hsa-mir-181c | dbDEMC |
| hsa-mir-23b | dbDEMC; HMDD | hsa-mir-224 | dbDEMC; HMDD |
| hsa-mir-337 | dbDEMC | hsa-mir-663 | dbDEMC; miR2Disease |
| hsa-mir-330 | dbDEMC | hsa-mir-190 | Unconfirmed |
| hsa-mir-346 | dbDEMC | hsa-mir-217 | dbDEMC |
| hsa-mir-150 | dbDEMC | hsa-mir-520c | miR2Disease; HMDD |
| hsa-mir-376b | dbDEMC | hsa-mir-612 | dbDEMC |

*Mitra et al. (2011)* that patients with breast cancer who had a poor prognosis is because of the low let-7b expression. *Ma et al. (2014)* discovered that miR-223 could make TNBCSCs more susceptible to induce TRAIL apoptosis *via* suppression of Hax-1. The above studies show that significant contribution to the expression of miRNAs in analyzing the clinicopathological characters of tumor tissues from breast cancer as well as for treating cancers. To identify possible miRNAs linked with breast tumors, we employ TLHNICMDA. Then sort these potential miRNAs based on the prediction score of TLHNICMDA. Finally, the HMDD v2.0, dbDEMC, and miR2Diseases were used to validate the top 50 miRNAs that may be linked with breast cancers. Table 4 shows that the first 10 miRNAs related to breast cancers were sustained by at lowest one of miR2Disease, dbDEMC, and the HMDD v2.0 databases. A total of 45 pairs of the first 50 miRNAs associated with breast cancers were confirmed by at least one of the miR2Disease, dbDEMC, and HMDD v2.0.

**Table 5  Verification of the top 50 miRNAs associated with esophageal neoplasms predicted by the proposed model.**

| miRNA | Database | miRNA | Database |
|---|---|---|---|
| hsa-mir-181d | dbDEMC | hsa-mir-510 | dbDEMC |
| hsa-mir-448 | dbDEMC | hsa-mir-204 | Unconfirmed |
| hsa-mir-199a | dbDEMC and HMDD | hsa-mir-220 | Unconfirmed |
| hsa-mir-186 | dbDEMC | hsa-mir-23b | dbDEMC |
| hsa-mir-124 | dbDEMC | hsa-mir-337 | Unconfirmed |
| hsa-mir-1 | dbDEMC | hsa-mir-34a | dbDEMC and HMDD |
| hsa-mir-629 | Unconfirmed | hsa-let-7d | dbDEMC |
| hsa-mir-10b | dbDEMC | hsa-mir-330 | dbDEMC |
| hsa-mir-377 | dbDEMC | hsa-mir-150 | dbDEMC and HMDD |
| hsa-mir-769 | dbDEMC | hsa-mir-346 | dbDEMC |
| hsa-mir-146b | dbDEMC | hsa-mir-149 | dbDEMC |
| hsa-mir-367 | dbDEMC | hsa-mir-376b | dbDEMC |
| hsa-mir-181a | dbDEMC | hsa-mir-600 | Unconfirmed |
| hsa-mir-433 | dbDEMC | hsa-mir-499 | Unconfirmed |
| hsa-mir-154 | dbDEMC | hsa-mir-424 | dbDEMC |
| hsa-mir-211 | dbDEMC | hsa-mir-25 | dbDEMC and HMDD |
| hsa-mir-602 | dbDEMC | hsa-mir-451 | dbDEMC |
| hsa-mir-153 | dbDEMC | hsa-mir-128a | Unconfirmed |
| hsa-mir-539 | Unconfirmed | hsa-mir-34c | dbDEMC and HMDD |
| hsa-mir-658 | Unconfirmed | hsa-mir-362 | dbDEMC |
| hsa-mir-140 | dbDEMC | hsa-mir-583 | dbDEMC |
| hsa-mir-145 | dbDEMC and HMDD | hsa-mir-134 | dbDEMC |
| hsa-mir-431 | dbDEMC | hsa-mir-200a | dbDEMC and HMDD |
| hsa-mir-136 | dbDEMC | hsa-mir-501 | dbDEMC |
| hsa-mir-421 | dbDEMC | hsa-mir-135b | dbDEMC and HMDD |

In recent years, the incidence of esophageal neoplasms has increased significantly. In different countries, the incidence and death rates of esophageal tumors vary greatly. As a high-incidence area of esophageal cancer, about 150,000 individuals die of this cancer in China (*He et al., 2012a*). *Wen et al. (2016)* found that miRNA-7 overexpressed in TE-1 in tumorous esophageal cells could reduce chemosensitivity *via* modulation of EGFR signaling through nuclear translocation. Moreover, two types pathological of EC have been validated to be interacted with miR-495, miR-181d, miR-25, miR-335, and miR-7. Among those miRNAs, the extent of differentiation of EC is associated with miR-25 and miR-130b, and the association between the survival probability of esophageal tumorous patients and the expression degree of miR-103/107 is validated as adversely associated. In particular, miR-130b and miR-25 can be taken as the gene breakthrough point to therapy for esophageal cancer (*He et al., 2012a*). Another study showed that overexpression of miR-133b, miR-133a, and miR-145 in the esophagus caused cancerous tissue to develop in the esophagus (*Kano et al., 2010*). The above studies demonstrate the importance of using

miRNAs as markers for the diagnosis of esophageal neoplasms. We used TLHNICMDA to verify potential miRNA's relationship with esophageal neoplasms and sort these potential miRNAs according to the model's prediction score. The validated databases are the miR2Disease, dbDEMC, and HMDD v2.0 database. Table 5 demonstrates that nine of the first 10 miRNAs linked with esophageal cancers were confirmed by at least one of the validated databases. In the first 50 miRNAs, 41 miRNAs linked with esophageal cancers were sustained by at least one of the validated databases.

## DISCUSSION

In this article, we introduced the disease similarity network, the miRNA similarity network, and known miRNA-disease association to form a disease-miRNA network. We further introduced the circRNA data to form a disease-miRNA-circRNA heterogeneous network for identifying the possible disease-miRNA interaction. Within the constructed disease-miRNA-circRNA network, an update algorithm was employed to extract information and effectively recognize interactions between miRNA and disease. Use 5-fold cross-validation and LOOCV to confirm the model TLHNICMDA in this article. Compare the AUC values of TLHNICMDA with five other models that predict disease-miRNA associations. The evaluation results showed that in local LOOCV and global LOOCV, the AUC values of TLHNICMDA are 0.7774 and 0.8795, separately. For 5-fold cross-validation, the procession of calculating the AUC was iterated 100 times to obtain the average and standard deviation of AUC of 0.8777+/−0.0010. After a comprehensive comparison, TLHNICMDA has better prediction performance. To further evaluate the applicability of TLHNICMDA for human complex diseases and the performance of independent prediction, two types of case studies were used. The four complex human diseases were employed as case studies to evaluate the performance of TLHNICMDA. The results of this study showed that among the first 50 miRNAs predicted by TLHNICMDA that are related to these four important diseases, 40, 40, 45, and 41 are verified by disease-miRNA associations databases and experimental reports, respectively. In conclusion, TLHNICMDA can effectively identify the possible disease-miRNA interactions as well as obtain reliable results. More importantly, TLHNICMDA as a disease-miRNA association prediction model can bring important contributions to the diagnosis, treatment, prevention, and prognosis of diseases.

However, the limitations of the TLHNICMDA model still exist. The adjacency matrix for disease-miRNA associations and miRNA-circRNA interactions is sparse, and the obtained outcomes may influence the precision of the model. Furthermore, the construction of the circRNA similarity network heavily relies on the Gaussian interaction profile kernel similarity of circRNA, which is excessively dependent on miRNA-circRNA interactions, thus partially affecting accuracy. Therefore, a complete similarity construction can be carried out on the collected circRNA sequence information in future research. In local LOOCV, the AUC value of TLHNICMDA outperformed RLSMDA, MCMDA, and TLHNMDA. Therefore, in the future, more biological data will be integrated to construct a four-layer or five-layer heterogeneous network for predicting the potential miRNA-disease associations, such as protein-protein associations, lncRNA-

miRNA interactions, lncRNA-disease associations, and so on. Additionally, we have no condition to do biological experiments. In the future, we will provide the validated miRNA-disease association if we have the condition to do biological experiments.

## CONCLUSIONS

In this study, we introduced a novel computational model, TLHNICMDA, designed to predict the potential miRNA-disease relationships. TLHNICMDA integrated the disease similarity, the miRNA similarity, the circRNA similarity, the known circRNA-miRNA links, and the known miRNA-disease interactions. To evaluate its effectiveness, we conducted two different types of case studies and three different types of cross-validation. The outcomes of three cross-validations showed that TLHNICMDA outperformed three comparative methods—TLHNMDA, MCMDA, and RLSMDA. The case studies in kidney neoplasms, colon tumors, breast cancer, and esophageal neoplasms showed that TLHMOCMDA can identify effectively possible interactions between miRNAs and diseases.

### Funding

This research was supported by Natural Science Foundation of Jiangsu Province under Grant BK20220621, the Natural Science Fund Project of Colleges in Jiangsu Province 21KJB520030, the National Natural Science Foundation of China under Grant 62106145, 62206177, the Zhejiang Provincial Natural Science Foundation of China under Grant LQ22F020024, LY23F020007, LTY22F020003, the Zhejiang Provincial Education Department Y202248951, and the School-Level Scientific Research Project of Shaoxing University 2021LG012. The funders had no role in study design, data collection and analysis, decision to publish, or preparation of the manuscript.

### Grant Disclosures

The following grant information was disclosed by the authors:
Natural Science Foundation of Jiangsu Province: BK20220621.
Natural Science Fund Project of Colleges in Jiangsu Province: 21KJB520030.
National Natural Science Foundation of China: 62106145, 62206177.
Zhejiang Provincial Natural Science Foundation of China: LQ22F020024, LY23F020007 and LTY22F020003.
Zhejiang Provincial Education Department: Y202248951.
The School-Level Scientific Research Project of Shaoxing University: 2021LG012.

### Competing Interests

The authors declare that they have no competing interests.

### Author Contributions

- Jia Qu conceived and designed the experiments, authored or reviewed drafts of the article, and approved the final draft.

- Shuting Liu performed the experiments, analyzed the data, authored or reviewed drafts of the article, and approved the final draft.
- Han Li conceived and designed the experiments, performed the experiments, analyzed the data, prepared figures and/or tables, and approved the final draft.
- Jie Zhou analyzed the data, authored or reviewed drafts of the article, and approved the final draft.
- Zekang Bian performed the computation work, authored or reviewed drafts of the article, and approved the final draft.
- Zihao Song conceived and designed the experiments, performed the computation work, authored or reviewed drafts of the article, and approved the final draft.
- Zhibin Jiang analyzed the data, authored or reviewed drafts of the article, and approved the final draft.

### Data Availability

The code and datasets are available in the Supplemental File.

### Supplemental Information

Supplemental information for this article can be found online at http://dx.doi.org/10.7717/peerj-cs.2070#supplemental-information.

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
