# Peer review of "Three-layer heterogeneous network based on the integration of CircRNA information for MiRNA-disease association prediction"

_PeerJ Computer Science, doi:10.7717/peerj-cs.2070_

## Round 0.1 · original submission · Major Revisions

The reviewers have substantial concerns about this manuscript. The authors should provide point-to-point responses to address all the concerns and provide a revised manuscript with the revised parts being marked in different color.

Reviewer 1 ·

Basic reporting

The article is well-written in clear, professional English. The introduction provides sufficient background and context, with references to relevant prior literature. The aims and scope of the work are clearly defined. The article structure generally follows standard format with distinct sections for the introduction, methods, results, discussion, and conclusion.

Some suggestions:

- More context about circRNAs and their relevance to miRNAs and diseases could be provided in the introduction. The authors mention circRNAs briefly but do not sufficiently explain their importance and connection to the study.

- Some file names inside the supplementary data (HMDD v3.2 and HMDD v2.0) is labelled in Chinese and should be converted to English.

- The “README.txt” and “vocab_compound.txt” file in HMDD v2.0 is not readable.

Experimental design

The research aims to develop an effective computational model for predicting miRNA-disease associations, which is within the scope of the journal. The research gap and rationale are clearly described. Predicting miRNA-disease interactions computationally can overcome limitations of biological experiments. The investigation appears to follow technically rigorous methodology and ethical standards. The TLHNICMDA model and its implementation are explained in sufficient detail to replicate the methods.

Some suggestions:

- The raw data on miRNA-disease associations, miRNA-circRNA interactions, disease semantic similarity, etc seem to be from standard databases, but it would be good to include more details on how the specific datasets were obtained, filtered or processed before use in this work.

- Statistical significance testing should be applied to compare AUC values between models.

Validity of the findings

The study does not make unsupported claims of impact or novelty. Replication of prior models provides meaningful comparison. The underlying data sources are cited and seem appropriate. The data processing and analysis methods appear statistically sound. The conclusions focus on the performance of the model on the specific datasets used and are supported by the results. Claims about causative relationships are not made.

Some suggestions:

- The conclusions focus on performance of the model, but more discussion is needed on biological significance of predictions. What new insights does this provide into miRNA-disease mechanisms?

- The comparison between models relies on AUC values alone. Reporting additional evaluation metrics would provide a more complete picture.

- The case studies are limited to top predictions - it would be better to analyze full rankings and lower ranked predictions. More investigation is needed into model errors and limitations.

·

Basic reporting

The article has some novelty, and here are some suggestions:

1.Outdated References: The majority of the literature cited in the manuscript is from over five years ago, a practice unacceptable in the rapidly evolving field of bioinformatics. The lack of references to more recent studies diminishes the credibility of the article. Please incorporate literature published after 2020.

2.Lack of Depth in Discussion: The discussion is superficial, lacking any in-depth analysis or critical thinking. For instance, on what basis do you assert that circRNA data can serve as a "bridge"? Is there any empirical evidence to support your hypothesis, or is it merely speculation?

3.Data Quality and Completeness: The models are built upon existing databases (e.g., HMDD, starBase, MeSH). Any inaccuracies, biases, or incompleteness in these databases would propagate through the study. Moreover, the study relies on data available up to a certain point in time, and newer associations might not be considered.

4.Complex Network Interactions: The study considers miRNA-disease, miRNA-circRNA, and disease-circRNA interactions. However, biological systems are incredibly complex, and there may be additional layers of interaction (e.g., protein-protein interactions, genetic modifications) that the model does not account for.

5.Assumption-based Similarities: The calculation of various similarities (e.g., semantic, functional, GIP) involves several assumptions and predefined parameters (like decay factors, Gaussian kernel bandwidth). If these assumptions are not accurate or if the parameters are not optimized, the model's predictions could be affected.

6.Statistical Validation: While the model uses cross-validation, the robustness of the model should be tested using additional statistical methods. The reliance on AUC values alone may not fully represent the model's predictive capabilities. Other metrics, potential overfitting, or the model's performance in real-world scenarios should also be evaluated.

7.Iterative Update Algorithm: The algorithm assumes that updating the interaction scores in iterations will lead to convergence on a stable set of scores. However, this may not always be the case, and the algorithm could be sensitive to initial conditions or fall into local optima.

8.Generalizability: The model's applicability to other types of RNA or diseases not included in the initial databases is unknown. Its performance in broader or different biological contexts needs to be assessed.

9.Computational Complexity: The methods used, especially the iterative update algorithm, could be computationally intensive, limiting the scalability of the model to larger datasets or more complex networks.

10.Experimental Validation: Computational predictions need to be validated experimentally. The study seems to lack a direct experimental validation of the predicted miRNA-disease associations, which is crucial for practical applications.

11.Potential Bias in Integrated Similarity Measures: The study integrates various similarity measures to construct the disease and miRNA networks. However, there might be a bias if one type of similarity measure dominates others, influencing the final integrated similarity.

12.Lack of External Dataset Testing: The model should ideally be tested on external datasets not involved in the training process to truly test its predictive power and generalizability.

Addressing these limitations in future research could involve incorporating more diverse types of biological data, refining the computational algorithms, and conducting extensive experimental validations to confirm the predicted associations.

Experimental design

no comment

Validity of the findings

no comment

Additional comments

no comment

Reviewer 3 ·

Basic reporting

The study of "Three-layer heterogeneous network based on the integration of CircRNA information for MiRNA-disease association prediction" proposed a model for miRNA-disease associate prediction by combining circRNA information with an existing model TLHNMDA. However, there are some significant concerns need to be revised and improved.

Experimental design

Experiments are not well designed to show and explain their results.
In “Materials & Methods”, associations, interactions and similarity methods should be well organized.
Two disease semantic similarity model were mentioned, but which one did you use?
Why do you use integrated similarity for diseases and miRNAs, and only GIP similarity for circRNAs?
Three validation methods (global LOOCV, local LOOCV and 5-fold corss-validation) are used to evaluate model performance. But you don’t compare them and justify which one is better and/or how they help evaluate these models.
Two types of case studies were mentioned, HMDD v1.0 and HMDD v2.0. What’s the difference and why do you use these two types?
Some vague or incorrect expressions:
• Line 203, DAG should be “directed acyclic graph”.
• Line 284 and 360, no explanations for FM, IM and ID.
• Line 299, this should be rephrased.

Validity of the findings

Results of this study don’t show better performance compared with previous model TLHNMDA when checking Table 1. So the integration of circRNA information may not be helpful for predicting miRNA-disease association. Also, there’s no discussion/explanations of why you should add circRNA information. Overall, the study doesn’t solve/improve the proposed issues.
In “Results”, some sentences need to be fixed.
• Line 393-395, “Diverse from the global LOOCV……with the disease under investigation”
• Line 410, “In contrast, disease-miRNA is less related.”
• Line 413, what’s “other five”?
• Line 449, “two other”? You also mentioned HMDD v2.0
• Line 542, “circRNA data as a bridge to connect diseases and miRNAs”. But from Figure 2 and 3, I believe miRNAs is the bridge between circRNAs and diseases?
• Line 564, “procision” should be “precision”.

Additional comments

Writing should be improved because of the bad structures and wrong expressions. Some of them have been mentioned above.
For “Introduction”, too much intro for miRNAs; intro for some related works isn’t concise and even two (TLHNMDA, RLSMDA) of four models compared in the study weren’t mentioned; line 134, “In the same year”, not same 2016, it’s 2018; line 178, “40, 40, 45, and 41 were verified by at lowest one database in miR2Disease, dbDEMC and HDMM v2.0.” isn’t clear.
For “Materials & Methods”, intro for circRNAs should be moved to “Introduction”.

---

## Round 0.2 · accepted · Accept

Reviewers are satisfied with the revisions and suggest accepting this manuscript.

Reviewer 3 ·

Basic reporting

All my concerns have been well addressed. The manuscript is ready to be published.

Experimental design

The study has detailed and reasonable experimental design.

Validity of the findings

Results are good and clear.